- Projections of future hydrologic drought in a reservoir-regulated region: the role of
- 3 climate change and reservoir operation
- Shaokun He<sup>1,5</sup>, Sirui Sun<sup>2</sup>, Yanghe Liu<sup>3</sup>, Kebing Chen<sup>4\*</sup>, Lingling Zhu<sup>4</sup>, Yu Gong<sup>1\*</sup>
- State Key Lab. of Water Resources Engineering & Management, Wuhan University, Wuhan
- 430072, China.
- <sup>8</sup> Middle Changjiang River Bureau of Hydrology and Water Resources Survey, Bureau of
- Hydrology of Changjiang Water Resources Commission, Wuhan 430010, China.
- <sup>3</sup>Three Gorges Cascade Dispatch & Communication Center, China Yangtze Power Co., Ltd.,
- Yichang 443000, China
- <sup>4</sup>Bureau of Hydrology of Changjiang Water Resources Commission, Wuhan 430010, China.
- <sup>5</sup>Department of Physical Geography and Ecosystem Science, Lund University, Lund 223 62,
- 14 Sweden
- \*Corresponding authors: chenkb@whu.edu.cn; ygong@whu.edu.cn

17 18

15

#### Abstract

Future hydrological droughts in reservoir-regulated regions remain unclear due to the complex interactions between climate change and reservoir operation. Existing studies usually make simple empirical assumptions about historical reservoir operation patterns to explore the role of climate change and reservoir operation on hydrological drought without even considering the role of optimal reservoir operation policies. Here, we take the upper Hanjiang River basin (UHRB) in China as a typical example to project its future hydrological drought evolutions using various standard streamflow indices (i.e., SSI-1, SSI-3, and SSI-12) and to quantify the role of each relevant factor. A new LSTM+Reservoir that combines a long and short-term memory (LSTM)based hydrological model with a physics-guided LSTM reservoir model is used to perform future projections using the meteorological outputs of five bias-corrected global climate models (GCMs) under three shared socioeconomic CMIP6 pathways (SSP126, SSP370, and SSP585). The results indicate that future climate change over the UHRB tends to reduce natural streamflow and exacerbate hydrological droughts, especially in the far-future period (2071-2100) under the SSP585 scenario. The operation of Ankang reservoir can mitigate drought frequency, duration, and severity for short-term SSI-1 and SSI-3 but fails for long-term SSI-12. Additionally, optimal reservoir operating policies that aim to maximize hydropower generation and pow generation guarantee rate can well reconcile the trade-off between short-term hydrological drought and hydropower benefits, which underscores the necessity of future reservoir operation improvements.

## 1 Introduction

Hydrological droughts, characterized by abnormally low streamflow in rivers, have significant direct and indirect ramifications on hydrological, agricultural, and social-economic sectors, such as losses of crops and hydropower generation (Chiang et al., 2021; Ji et al., 2023;

Van Vliet et al., 2016). In the last decades, hydrological droughts have become more frequent in the Americas, East Asia, and Africa, and global warming arising from high greenhouse gas concentrations has been identified as the main driver (Gudmundsson et al., 2021). According to the Sixth Assessment Report (AR6) of the Intergovernmental Panel on Climate Change (Ipcc, 2021), the time series of land temperatures is projected to continue to rise, which will inevitably exacerbate extreme hydrological droughts in a warming future. Hence, it is of great importance to assess the characteristics of extreme hydrological drought in the context of climate change to enable effective adaptation strategies. In the meantime, the rapid expansion of reservoirs, a typical element of human activities, has created new challenges for the assessment of future hydrological droughts. To date, there are more than 55,000 reservoirs worldwide registered by the International Commission on Large Dams, with a total storage capacity of 14,602 km<sup>3</sup> (Eriyagama et al., 2020). The large capacity of so many reservoirs can dramatically alter drought patterns by regulating the spatiotemporal distribution of river flows (Brunner et al., 2021; Chang et al., 2019). Wanders and Wada (2015) suggest that the dampening effect of reservoirs on the majority of strongly regulated river basins in Europe and North America, relative to the natural climate change scenario, would help to reduce drought severity during low-flow seasons. Wan et al. (2018) reported that irrigation reservoirs would intensify the duration and intensity of global hydrological droughts by 50% over the period 2070– 2099. It can be argued that investigating the impact of reservoir operation on future hydrological droughts is region-dependent due to the various functions of reservoirs (e.g., hydropower generation, irrigation, and flood control) as well as the heterogeneity of regional climate change. Recently, some scholars have begun such drought analysis efforts in some key watersheds (Sun et al., 2023; Xing et al., 2021; Wu et al., 2023). For example, Yun et al. (2021b) attempted

to assess the effectiveness of reservoir operation in modifying hydrological extremes in the Lancang-Mekong River basin using five global climate models (GCMs) from the sixth Coupled Model Intercomparison Project (CMIP6) and the VIC-Reservoir model. Ji et al. (2023) projected hydrological drought changes in the upper Yellow River basin under different levels of global warming by forcing a hybrid Conjunctive Surface-Subsurface Process Version 2 (CSSPV2) and reservoir model with bias-corrected CMIP6 meteorological forcing data. These drought experiments demonstrated the availability of coupling hydrological and reservoir modules for such problems, but they might draw inaccurate conclusions from empirical assumptions about how reservoirs were operated without considering actual reservoir operation data. In fact, critical historical operating data contain rich decision-making information reflecting sophisticated anthropogenic operational behaviors across multiple inflow scenarios (Zheng et al., 2022). A state-of-the-art tool that can scientifically mine massive historical operating data is critical for capturing reservoir releases associated with hydrological droughts.

In order to overcome these limitations, machine learning (ML) is considered a promising alternative to reproduce historical reservoir operation processes due to its continuous successful application in hydrological time series simulation. Artificial neural network (ANN) (Özdoğan-Sarıkoç et al., 2023), nonlinear autoregressive models with exogenous input (NARX) (Yang et al., 2019) and long short-term memory (LSTM) (Zhang et al., 2018) have been used to implement historical reservoir operation simulation using large-sample data. LSTM in particular can achieve favorable results, while embedding physical mechanisms into it can even further enhance its understanding of operational behaviors under extreme hydrological conditions, allowing for more accurate high- and low-flow simulations (Zheng et al., 2022). In addition, the original physical hydrological models (e.g., VIC and CSSPV2) used in future drought analysis can also be replaced

by powerful LSTMs, which, once interfaced with a LSTM-based reservoir operation module, can trigger a fully artificial intelligence (AI)-based simulation, providing new insights into the automated diagnosis of future hydrology droughts.

Beyond exploring the impact of historical operating policies on future hydrological droughts, it is also crucial to explore how effective optimal operating policies are in the trade-off between operating benefits and hydrological extremes. As one of the nature-based solutions (NBSs), optimal reservoir operation is favored to boost water resource benefits without additional capital investment (He et al., 2022; Zolfagharpour et al., 2021). Most of the existing literature solely makes a theoretical call to consider hydrological drought when pursuing optimal operating benefits and vice versa, but without any substantive implementation analysis (Chang et al., 2019; Ji et al., 2023; Wu et al., 2023). It remains unclear whether incorporating such optimal strategies into current water management practices will improve or deteriorate basin resilience to hydrological drought extremes under climate change.

Here, we investigate how much climate change and reservoir operation influence future hydrological droughts under three CMIP6 shared socioeconomic pathways, using the upper Hanjiang River basin in China, a heavily reservoir-regulated region, as a typical example. To explicitly quantify the role of climate-induced and reservoir-induced factors on hydrological droughts, we first construct a LSTM-based hydrological model and physics-guided LSTM reservoir model to simulate historical reservoir inflow and outflow, respectively. We then drive the hybrid LSTM model with the outputs of five bias-corrected CMIP6 GCMs under three future scenarios to project daily river streamflow for near-future and far-future periods. Subsequently, different drought characteristics (e.g., duration, frequency, severity) are extracted from the run theory and analyzed under both past and future climates. Finally, we assess the impact of optimal

operating policies on future hydrological droughts and compare them against the historical policy to highlight their advantages in reconciling hydrological droughts with operating benefits.

## 2 Study Area and Data Description

### 2.1 Study area

The Hanjiang River basin in central China is the lifeblood of the water economy in the riparian provinces. As the longest tributary of the Yangtze River, the river has experienced a great deal of anthropogenic intervention in the construction of hydraulic infrastructure, including a series of reservoirs and inter-basin water transfer projects. Based on its geographical topography, the basin is usually divided into upper, middle and lower sections. In this study, the uppermost Hanjiang River basin (UHRB), which originates in the southern foothills of the Qinling Mountains and terminates at the Ankang hydrological station (shown in Figure 1), is used as a reference case. The UHRB is located at 31°-34.5°N latitude and 106°-109.5°E longitude, with a subtropical monsoon climate. The average annual precipitation, temperature and streamflow are about 850mm, 15 °C, and 500mm, respectively. The flood season (May to October) accounts for 75% of the total annual precipitation. Runoff has a similar temporal distribution, which makes the UHRB vulnerable to natural disasters such as floods and droughts (Jin et al., 2023). There is an urgent need for effective water resource management and disaster preparedness measures in the UHRB. The Ankang reservoir, with a total capacity of 3.2 billion m<sup>3</sup>, is one such engineering target, as it is the largest and last reservoir in the basin. Since its commissioning in 1990, the reservoir has been used primarily for hydropower generation (installed capacity: 850 MW), supplemented by flood control, navigation and other functions (Chinese National Committee on Large (Dams, 2011). The reservoir has a natural catchment area of nearly 35,700 km<sup>2</sup> and an active storage capacity of 1.47 billion m<sup>3</sup>, which is used to regulate the spatially unevenly distributed reservoir

flows. Flow monitoring devices have been installed at the entrance to the reservoir and nearly 30 km downstream (i.e., at the Ankang Hydrological Station) to measure inflows and outflows, respectively.

**Figure 1.** Location of the upper Hanjiang River Basin (UHRB) and distribution of major hydrological elements, including hydro-meteorological stations and Ankang reservoir.

2.2 Data

The research dataset used in this study includes both historical in-situ observations and future projections. Historical meteorological records of 11 meteorological stations in Figure 1 for the period 1992–2020 are archived at the China Meteorological Administration Data Sharing Service Center (CMA, http://data.cma.cn), which include daily precipitation (Pr, mm), wind speed (Win, m/s), relative humidity (Rh, %), air temperature (maximum, minimum, and average Tem, °C),

167168

and so on. Time series of basin-averaged precipitation and temperature are derived using the Thiessen polygon method. Observed streamflow data for the same historical period are obtained from the Bureau of Hydrology of the Yangtze Water Resources Commission of China (https://www.cjh.com.cn), where Ankang reservoir inflow can be regarded as a near-natural flow without anthropogenic disturbance.

For future climate projections, we used a multi-model ensemble that includes the five GCMs in Table 1 with three shared socioeconomic pathways (i.e., SSP126, SSP370, and SSP585) from the latest CMIP6. Numerous studies have found that the raw climate data (e.g., precipitation, air temperature) in CMIP6 are overestimated in Asia with non-negligible uncertainties (Chai et al., 2022). To reduce the systematic biases of climate models, we use the bias-corrected daily output the Inter-sectoral Impact Model Intercomparison Project 3b (ISIMIP3b, https://data.isimip.org/search/tree/ISIMIP3b/InputData/), which has been downscaled to a spatial resolution of  $0.5^{\circ} \times 0.5^{\circ}$  by using climate observations from 1850 to 2100. In the bias adjustment process of the ISIMIP3b, Lange (2019) used a trend-preserving parametric quantile mapping method and took interdependencies between different variables into account, thus providing significant advantages over its predecessor (i.e., ISIMIP2). This dataset has demonstrated the robustness of its performance in many regions of China (Kang et al., 2023; Yun et al., 2021a; He et al., 2023). To assess climate change impacts, three equal 30-year periods were defined as the reference (1985–2014), near-future (2031–2060), and far-future (2071–2100) periods.

Table 1. Basic information on the five global climate models (GCMs) from IMISIP3b

| ID | Model            | Group) | Institution Name                          | Horizontal resolution (lon. × lat.) |
|----|------------------|--------|-------------------------------------------|-------------------------------------|
| 1  | IPSL-CM6A-<br>LR | IPSL   | Institute Pierre Simon Laplace,<br>France | $2.50^{\circ} \times 1.27^{\circ}$  |

| 2 | GFDL-ESM4         | NOAA-<br>GFDL | Geophysical Fluid Dynamics<br>Laboratory, Princeton                         | 1.25° × 1°                           |
|---|-------------------|---------------|-----------------------------------------------------------------------------|--------------------------------------|
| 3 | MPI-ESM1-2-<br>HR | MPI-M         | Max Planck Institute for<br>Meteorology, Germany                            | $0.9^{\circ} \times 0.9^{\circ}$     |
| 4 | MRI-ESM2-0        | MRI           | Meteorological Research Institute,<br>Japan                                 | $1.125^{\circ} \times 1.125^{\circ}$ |
| 5 | UKESM1-0-LL       | MOHC<br>NERC  | Met Office Hadley Centre and<br>Natural Environment Research<br>Council, UK | 1.25° × 1.875°                       |

# 3 Methodology

This section gives a methodology for exploring future hydrological droughts under the coupled effects of climate change and reservoir operation, as shown in Figure 2. First, we perform a LSTM-based reservoir inflow simulation and a physics-based LSTM reservoir operation simulation. Then, the ISIMIP3b outputs are used to drive the hybrid model to project future streamflow scenarios and extract hydrological drought characteristics. Finally, several different experiments are designed to investigate the individual roles of climate change and reservoir operation in future hydrological droughts. Each of these modules is described in the following subsections.

**Figure 2.** Schematic diagram of the modules used in this study to explore the role of climate change and reservoir operation in future hydrological droughts. The meanings of the acronyms in the experimental description panel are given in Section 3.3 below.

## 3.1 Long short-term memory (LSTM)

The LSTM is an variant of recurrent neural network that uses the backpropagation through time (BPTT) method to get around the issue of vanishing gradients and keep track of information from earlier time steps (Hochreiter and Schmidhuber, 1997). It is specially structured with a productive memory block to replace the hidden layer nodes of conventional neural networks (Hochreiter, 1998; He et al., 2022). The memory block (shown in Figure 3a) consists of a forget gate, an input gate, an output gate, and a memory cell. The forget gate decides which information from the previous cell state is to be discarded, whereas the input gate determines what information is important enough to update the cell state. The output gate uses the cell state to generate the value of the next hidden state. Mathematically, a typical memory block of LSTM can be described by the following Equations (1) to (5).

195 
$$f_{t} = \sigma(x_{t}W_{f} + h_{t-1}U_{f} + b_{f})$$
 (1)

196 
$$i_t = \sigma(x_t W_i + h_{t-1} U_i + b_i)$$
 (2)

197 
$$o_{t} = \sigma(x_{t}W_{o} + h_{t-1}U_{o} + b_{o})$$
 (3)

198 
$$c_{t} = f_{t} \otimes c_{t-1} + i_{t} \otimes \tanh(x_{t}W_{c} + h_{t-1}U_{c} + b_{c})$$
 (4)

$$199 h_{t} = o_{t} \otimes \tanh(c_{t}) (5)$$

where  $x_t$ ,  $f_t$ ,  $i_t$ , and  $o_t$  are input variables, forget gate, input gate, and output gate at time t, respectively.  $c_t$  and  $h_t$  are the cell state and the hidden state at time t, respectively, while  $c_{t-1}$  and  $h_{t-1}$  are at the previous time t-1. W, U and b with various subscripts denote input weights, recurrent weights and bias terms, respectively.  $\sigma(\cdot)$  is the sigmoid activation function with a return value from 0 to 1.  $\tanh(\cdot)$  is the hyperbolic tangent activation function with a return value from -1 to 1.  $\otimes$  is the element-wise multiplication.

# (c) sequence-to-sequence

**Figure 3.** Model structure of long and short-term memory (LSTM). (a) The internal structure of a standard LSTM memory block, consisting of a forget gate, an input gate, an output gate, and a memory cell. (b) A three-layer sequence-to-one LSTM structure modelled by correlated meteorological inputs to simulate reservoir inflow. (c) A physics-guided LSTM-based sequence-to-sequence model with inputs of antecedent reservoir storage, time of year (*toy*), precipitation and simulated reservoir inflow to simulate reservoir outflow, where a red block following the LSTM block represents a set of operational constraints, including the water balance equation and reservoir storage and outflow limits.

### 3.1.1 LSTM-based reservoir inflow simulation

Some hydrological experiments have shown that a three-layered LSTM with one hidden depth is robust and high-precision enough to reflect the nonlinear rainfall-runoff relationship, although its black-box nature makes the interpretation of physical processes more challenging (Konapala et al., 2020; Liu et al., 2022; Rehana and Rajesh, 2023). This topology is adopted in this study (Figure 3b), where relevant meteorological variables (mainly Pr and Tem) with certain lag times are considered inputs and near-natural reservoir inflow at the current time is considered the output. The lag time is determined by the cross-correlation coefficient method (Cui et al.,

2022). In addition, the number of hidden units and the initial learning rate are significant LSTM hyper-parameters to be determined. More details of the hyper-parameter tuning process are provided in Table S1 of Supporting Information, which is implemented by the Keras module in the backend of Python's *Tensorflow* module (Abadi et al., 2016). Notably, antecedent reservoir inflow is not used as an input variable in this study, although it is closely related to the model output in reality. This is because it is not possible to accurately predict antecedent reservoir inflow under future scenarios, which can only be inferred through model simulation. The inclusion of antecedent reservoir inflow in the input pool may result in a gradual widening of the model simulation error. For historical simulations, the meteorological data from 1992 to 2020 were used, with 1992 reserved as the model spin-up period, and the remaining data split into a calibration period (1993–2014) and a validation period (2015–2020) according to an 80%–20% rule. For future projections, the period 1985–2100 is used to cover the simulation span of the three SSP scenarios. Within this range, the period 1985-2014 is designated as the reference period (following ISIMIP3b protocol) to evaluate future streamflow variations against a consistent historical baseline. The LSTM model is trained on 1992-2020 due to observational data availability, while the non-overlapping reference period is used independently for climate impact assessment. 3.1.2 Derivation of historical operation patterns with a physics-guided LSTM model For human-intervened reservoir operation modules involving a great deal of expertise, LSTM, as one of the state-of-the-art data mining techniques, is still more accurate than the traditional hypothetical empirical equations in extracting operation policies from massive historical records (Zheng et al., 2022; Longyang and Zeng, 2023; García-Feal et al., 2022). Similar to Figure 3(b),

we constructed a three-layer sequence-to-sequence LSTM model (Figure 3(c)) to simulate the

264

266

267

268

269

reservoir outflow sequence. Following the guidelines from the local reservoir management agency, we used antecedent reservoir storage state, time of year, precipitation, and reservoir inflow as the major inputs. It is worth noting that to ensure the robustness of the model for future simulations, we used the LSTM outputs in Section 3.1.1 instead of actual inflow observations. In addition, a new hidden state of reservoir storage, initially set to be the flood-limited water level, is designed in the LSTM structure, as it can be determined by a state transition equation (i.e., the water balance equation in Equation (6)). In order to avoid anomalies during the simulation (e.g., violation of realistic reservoir physical properties), other operational constraints (i.e., reservoir storage limits in Equation (7) and reservoir outflow limits in Equation (8)) are also integrated into the LSTM, culminating in its physics-guided variant.

$$V_{t+1} = V_t + (I_t - O_t) \cdot \Delta t \tag{6}$$

$$V_{min} \le V_t \le V_{max} \tag{7}$$

$$O_{\min} \le O_t \le O_{\max} \tag{8}$$

where  $V_t$  and  $V_{t+1}$  are the initial and terminal reservoir storage (m<sup>3</sup>) at time t, respectively;  $I_t$  and 259  $O_t$  are reservoir inflow (m<sup>3</sup>/s) and outflow (m<sup>3</sup>/s) at time t, respectively;  $V_{min}$  and  $V_{max}$  are the 260 allowable minimum and maximum reservoir storage (m<sup>3</sup>), respectively;  $O_{min}$  and  $O_{max}$  are the 261 allowable minimum and maximum reservoir release (m<sup>3</sup>/s), respectively; and  $\Delta t$  is the time step 262 (s) of the simulation period. 263

265

### 3.1.3 Evaluation metrics for model performance

To simultaneously ensure the simulation accuracy of near-natural reservoir inflow and human-regulated outflow, we chose the average of their respective Nash-Sutcliffe efficiency (NSE, Equations (9)–(10)) as the optimization objective and the adaptive moment estimation (ADAM) algorithm (Kingma and Ba, 2014) as the optimization method.

270 
$$\max NSE_{ave} = 1/2 \times (NSE_{inflow} + NSE_{outflow})$$
 (9)

271 
$$NSE_{i} = 1 - \sum_{t=1}^{T} (Q_{t}^{i,sim} - Q_{t}^{i,obs})^{2} / \sum_{t=1}^{T} (Q_{t}^{i,obs} - \overline{Q^{i,obs}})^{2}$$
 (10)

where  $Q_t^{i,sim}$  and  $Q_t^{i,obs}$  denote the simulated and observed streamflow, respectively, for the *i*th series (either inflow or outflow) at time t;  $\overline{Q^{i,obs}}$  is the mean observed streamflow for the *i*th series; and T is the total number of time periods. The NSE is a widely used performance metric for hydrological modeling, with values ranging from  $-\infty$  to 1, where 1 indicates a perfect match between simulation and observation.

### 3.2 Standard streamflow index

This study used the standardized streamflow index (SSI) to describe hydrological drought because it only needs streamflow data and has been shown to work across a range of timescales, including 1, 3, 12, and 24 months (Vicente-Serrano et al., 2012; Smith et al., 2019; Gu et al., 2020; Shukla and Wood, 2008). The 1-month (SSI-1) and 3-month (SSI-3) scales of SSI show short-term wet/dry hydrological conditions. The 12-month (SSI-12) and 24-month (SSI-24) scales describe cumulative streamflow anomalies over 12 and 24 consecutive months, respectively, which show long-term drought conditions. Here, SSI-1, SSI-3, and SSI-12 were selected to reflect the monthly, seasonal, and annual hydrological drought, respectively.

In the calculation of SSI for each calendar month m (m = 1, 2,..., 12) at a specific time scale, a Pearson type-III distribution with the Kolmogorov-Smirnov test is first used to fit the corresponding streamflow series (Q) during the reference period.

289 
$$F_{m}(Q) = \frac{\beta^{\alpha}}{\Gamma(\alpha)} \int_{\gamma}^{\infty} (Q - \omega)^{\alpha - 1} e^{-\beta(Q - \omega)} dr$$
 (11)

where  $F_m(Q)$  is the cumulative distribution function;  $\alpha$ ,  $\beta$ , and  $\omega$  are the shape, scale, and location parameters of the distribution, which can be estimated by the L-moment method (Hosking, 1990). The SSI values for the reference period can be obtained by a standard normal transforming process  $(\Phi^{-1})$ .

 $SSI = \Phi^{-1}(F_m)$ (12)The same distribution parameters derived from the 30-year reference period were applied to the two 30-year future periods (i.e., the near-future and far-future) to assess future SSI changes, ensuring consistency in comparing hydrological droughts (either climate-induced or reservoirinduced) between future simulations and historical baselines (Yun et al., 2021b; Wan et al., 2018). The characteristics (e.g., duration, severity, and intensity) of hydrological drought episodes were extracted using the run theory (Yevjevich, 1967). A hydrological drought episode starts when the SSI value falls below a threshold (-0.5), and ends when the SSI value recovers above the threshold, as in the case of the two drought episodes  $D_0$  and  $D_1$  in Figure 4. Drought duration is defined as the length of a drought episode, severity as the cumulative deficit of SSI values below the drought threshold during the episode, and intensity as the average deficit below the threshold over the episode, calculated as severity divided by duration. In particular, the pair  $d_0$  and  $d_2$  can be merged into a single drought episode (i.e., the third drought episode in Figure 4) when the time interval d1 between two adjacent drought branches is no longer than the time evaluation criterion  $t_c$  ( $t_c = 2$  months in this study) and the SSI during  $d_1$  remains below the allowable upper threshold (Zhou et al., 2021; Wu et al., 2017). The corresponding duration for this merged episode is  $D_2 = d_0$  $+d_1+d_2$  and the severity is  $S_2 = s_0 + s_2$ . Since drought intensity is defined as the ratio of severity to duration, only two drought characteristics, duration (D) and severity (S), are used in this study to comprehensively characterize each drought episode.

**Figure 4.** Identification of hydrological drought events and characteristics using run theory. Three types of drought episodes are illustrated in orange: episode  $D_0$  with severity  $S_0$ , episode  $D_1$  with severity  $S_1$ , and a merged episode  $D_2$  with severity  $S_2$ , where  $D_2 = d_0 + d_1 + d_2$  and  $S_2 = s_0 + s_2$ . The pair  $d_0$  and  $d_2$  is merged into a single drought episode since the interval  $d_1$  between these two adjacent branches no longer than the time evaluation criterion  $t_c$  and the SSI remains below the upper threshold during this interval.

### 3.3 Experimental Design

To fully explore the role of climate change and reservoir operation in future hydrological droughts, several numerical experiments are designed in Table 2. OBS/LSTM and OBS/LSTM+Reservoir denote simulations forced by observed CMA meteorology in the absence and presence of reservoir operation, respectively. ISIMIP3b\_ref/LSTM and ISIMIP3b\_ref+Reservoir are similar but forced by ISIMIP3b forcings during the reference period. ISIMIP3b\_fut/LSTM and ISIMIP3b\_fut/LSTM+Reservoir progressively account for the impacts of climate change and reservoir operation on future projections. Notably, the symbol "Reservoir" in the experiment refers to the historical reservoir operation policy for the period 1992–2020 derived from the physics-guided LSTM model.

There is little focus on the evolution of trade-offs between operating benefits and hydrological drought risk, although a large body of literature points out the necessity of optimizing

reservoir operation policies (Ji et al., 2023; Brunner, 2021; Wu et al., 2022; Firoz et al., 2018). To this end, a classical multi-objective decision-making optimization is implemented for the Ankang Reservoir to maximize both hydropower generation and the power generation guarantee rate. The optimal set of alternative operating policies  $\pi_{\theta}^*$  over the historical climate conditions  $w^H$  can be yielded by solving the following problem.

338 
$$\pi_{\theta}^* = \arg \max_{\pi_{\tau}} \mathbf{f}(\pi_{\theta}, w^H) = |f_{THP}(\pi_{\theta}, w^H), f_{PGR}(\pi_{\theta}, w^H)|$$
 (13)

where  ${\bf f}$  is the objective vector of [  $f_{THP}$ ,  $f_{PGR}$  ] (refer to Text S2 for more details). The policies  $\pi_{\theta}$ , parameterized as Gaussian radial basis functions, a formulation shown to be effective for reservoir operation optimization (Quinn et al., 2019; Bertoni et al., 2019). Optimization was performed using Non-dominated Sorting Genetic Algorithm II (NSGA-II; (Deb et al., 2002). The resulting Pareto-optimal policies,  $\pi_{\theta}^*$ , were then applied in future climate scenarios to explore the potential co-benefits and trade-offs between hydropower generation and drought risk reduction. This exploratory analysis is represented by the ISIMIP3b\_fut/LSTM+Reservoir\_Opt experiment in Table 2, where detailed in-depth analysis is provided.

**Table 2.** Experimental design and scenario configurations used in this study.

| Experiment                  | Meteorological forcing | Simulation period       | Climate change | Traditional reservoir operation | Optimal reservoir operation |
|-----------------------------|------------------------|-------------------------|----------------|---------------------------------|-----------------------------|
| OBS/LSTM                    | Observations           | 1992-2020               | _              | _                               | _                           |
| OBS/LSTM + Reservoir        | Observations           | 1992-2020               | _              | ✓                               | _                           |
| ISIMIP3b_ref/LSTM           | ISIMIP3b reference     | 1985–2014               | -              | _                               | _                           |
| ISIMIP3b_ref/LSTM+Reservoir | ISIMIP3b reference     | 1985–2014               | _              | ✓                               | _                           |
| ISIMIP3b_fut/LSTM           | ISIMIP3b<br>future     | 2031–2060,<br>2071–2100 | ✓              | _                               | _                           |
| ISIMIP3b_fut/LSTM+Reservoir | ISIMIP3b future        | 2031–2060,<br>2071–2100 | <b>√</b>       | ✓                               | _                           |

| ISIMIP3b fut/LSTM+Reservoir Opt  | ISIMIP3b | 2031–2060, | ✓ - |   | ,        |
|----------------------------------|----------|------------|-----|---|----------|
| isiwiipsu_iu/LsTwi+Reservoii_Opt | future   | 2071-2100  |     | _ | <b>V</b> |

### **4 Results and Discussion**

## 4.1 Model calibration and validation

Figure 5 presents the calibration and validation results for both reservoir inflow and outflow using the LSTM-based modeling framework. As shown in Figure 5(a), the LSTM model can accurately simulate near-natural reservoir inflow, particularly at the monthly scale. The *NSE* values for the calibration and validation periods reach 0.95 and 0.93, respectively, exceeding the widely accepted performance threshold (*NSE* > 0.5) for hydrological modeling (Moriasi et al., 2007). Figure 5(b) illustrates the comparison between observed and simulated reservoir outflows at the Ankang hydrological station. The seasonal shift between observed inflow and outflow curves (black lines in Figure 5(a) and 5(b)) suggests that reservoir operations have reshaped streamflow seasonality, with an estimated 5–21% of downstream flow withheld by the Ankang reservoir during June–October and released later in the year. This operational pattern is well captured by the LSTM+Reservoir model driven by observed meteorological forcings, yielding *NSE* values of 0.91 and 0.89 for the calibration and validation periods, respectively. While slightly lower than those for inflow, these values reflect satisfactory performance given the complexity of human-influenced reservoir operation.

Figure 5 also shows the ensemble-averaged hydrographs from the ISIMIP3b\_ref/LSTM and ISIMIP3b\_ref/LSTM+Reservoir experiments, driven by ISIMIP3b meteorological forcings instead of historical meteorological observations. The model performance under these forcings is noticeably weaker than that of the OBS/LSTM and OBS/LSTM+Reservoir configurations, likely due to the limited ability of ISIMIP3b in characterizing regional-scale meteorological regimes

(Kang et al., 2023). Nevertheless, the simulated low-flow conditions of ISIMIP3b\_ref align closely
with observations in both magnitude and duration, providing a reliable basis for the subsequent
hydrological drought analysis.

**Figure 5.** Hydrographs of (a) near-natural reservoir inflow (without reservoir operation) and (b) reservoir outflow. Simulations driven by meteorological observations (i.e., OBS/LSTM and OBS/LSTM+Reservoir experiments) are marked as blue lines. The ensemble mean and  $\pm 1$  standard deviation of simulations driven by ISIMIP3b GCM meteorological data (i.e., ISIMIP3b/LSTM and ISIMIP3b/LSTM+Reservoir experiments) are marked as orange lines and shaded bands, respectively.

Changes in reservoir storage ( $\Delta S$ ) represent another key variable in our operation simulations (used in the hydropower performance assessment presented in Section 4.4). Figure 6 illustrates the observed and simulated monthly mean storage variations over the available period 2001–2010.

Both the OBS/LSTM+Reservoir and ISIMIP3b\_ref/LSTM+Reservoir simulations reproduce the observed dynamics well, particularly the storage accumulation from July to November. With the correlation coefficients between simulated and observed storage series ranging from 0.70 to 0.73, the model provides a reasonable approximation of reservoir operations and is suitable for subsequent analysis.

**Figure 6.** Change in average monthly storage ( $\Delta S$ ) in the Ankang reservoir during 2001-2010. The black dotted line represents multi-year observations. The blue line shows simulation results from the OBS/LSTM+Reservoir simulation. The orange boxplots represent the ISIMIP3b/LSTM+Reservoir ensemble simulations driven by five GCMs from ISIMIP3b.

# 4.2 Streamflow variation under the impacts of climate change and reservoir operation

Climate change scenarios in ISIMIP3b project a consistent upward trend in both precipitation and temperature over the UHRB during the future periods, relative to the reference period (at a significance level of p

+11.2%) but a more pronounced warming (+1.8°C to +4.0°C). SSP585 is projected to experience the largest increases in both precipitation (+8.0% to +15.8%) and temperature rise (+2.3°C to +5.3°C). As a result of the combined effects of these two major climatic drivers, the multi-year average reservoir inflow is expected to increase from +0.3% (near-future, 2031–2060) to +5.5% (far-future, 2071–2100) under the SSP126 scenario. Under SSP370 and SSP585, it is expected to shift from +0.2% (near-future) to -7.0% (far-future), and from -2.6% (near-future) to -8.4% (far-future), respectively, suggesting a potential long-term decline despite short-term gains. This implies that warming-induced evaporation losses may outweigh the stimulatory effects of increased precipitation, especially under higher-emission scenarios (Satoh et al., 2022).

Figure 7 further illustrates the projected relative change in monthly average streamflow under different future periods and SSP scenarios, explicitly highlighting the seasonal influence of both climate change and reservoir operation. Substantial inter-model uncertainty is evident, particularly under SSP585 during the far-future flood season, where streamflow changes range from -45% to +43%. Despite this variability, the ensemble mean reveals a consistent signal: positive deviations are largely concentrated in the flood season, while most other months are expected to experience declining streamflow. This asymmetric seasonal response suggests a likely intensification of hydrological seasonality, with wetter periods becoming more prone to floods and drier periods experiencing heightened water stress. Human-regulated reservoir operation has the potential to moderate the magnitude of future monthly streamflow changes. However, across all scenarios, we find that the extent to which the Ankang Reservoir alters streamflow patterns remains rather limited. This may be attributable to the reservoir's primary operational objective of hydropower generation, with relatively little emphasis placed on shaping the flow regime itself. Therefore, further investigation into effective reservoir management is warranted.

Therefore, further investigation into effective reservoir management is warranted.

**Figure 7.** Relative changes in projected monthly streamflow for two future periods and three SSP scenarios under the impacts of climate change and reservoir operation relative to the reference period (1985–2014). Lines are the ensemble mean of the five GCMs, and areas represent the uncertainty of the five GCMs.

## 4.3 Changes in hydrological drought events

To comprehensively evaluate future hydrological droughts, we analyzed both the continuous SSI-based drought characteristics and the annual drought event frequency and severity under different climate and reservoir operation scenarios. The time series of SSI-3 associated with reservoir inflow and outflow, along with their ensemble spreads under three emission scenarios, are shown in Figure 8, with SSI-1 and SSI-12 counterparts in Figures S1 and S2, respectively. SSI-1 and SSI-3 exhibit substantial intra-annual fluctuations within [-3, 3], whereas SSI-12 displays smoother variability reflecting more stable dynamics. Consistent with projected reductions in streamflow, all three indices (SSI-1, SSI-3, and SSI-12) show a slight worsening trend over time, particularly under SSP370 and SSP585, indicating an increased likelihood of drought occurrence in the future (Figures 8(b), 8(d) and 8(f)). We therefore counted the number of drought events for the three different periods estimated by the GCMs and visualized them on the right side of Figures

8, S1 and S2. These subplots show that the number of drought events in the future period is higher than that in the reference period, despite the large discrepancy in the estimates from different GCMs. The number of drought events in the near-future period is slightly higher than that in the far-future period, with more small, frequent droughts. In addition, as shown in Figure 8 (b), (d), and (f), reservoir operation can mitigate the frequency of drought in the reference period but does not completely remove the risk of hydrological drought under future climate change. Reservoir operation is better at preventing short-term droughts, as the drop in the number of droughts associated with reservoir release versus inflow is significant for SSI-1 in Figure S1 but not for SSI-12 in Figure S2. It may be due to the inadequate annual regulation capacity of the Ankang reservoir.

A comprehensive assessment of SSI-3 drought characteristics, including duration and severity, is further given in Figure 9 (see Figure S3 and Figure S4 for SSI-1 and SSI-12, respectively). Drought duration and severity in this basin are expected to deteriorate due to climate change. The extreme hydrological drought associated with SSI-3 is projected to occur in the far-future period under the SSP585 scenario, with a maximum duration of 33 months and a maximum severity of 47.8. It will then be followed by the scenario SSP370, with an 18-month duration and a severity of 22.9, and finally the SSP126 scenario, with a 12-month duration and a severity of 12.4. The drought duration and severity associated with SSI-1 and SSI-12 share a similar pattern. All indications are that SSP585 has the most profound impact on hydrological drought in the region. Notably, reservoir operation can provide significant relief from extreme hydrological drought pressure because it can divide more short-term drought events through reservoir impoundment and release regulation. For the far-future period under SS`P585, the maximum duration associated with SSI-3 is reduced by 72.73% and the maximum severity is reduced by 63.81% by reservoir

- operation. The original extreme hydrological drought associated with SSI-1 can likewise be
- regulated to a modest level, as Figure S3 shows. It is yet invalid for SSI-12, which requires more
- human activities to improve.

Figure 8. Hydrological drought SSI-3 for reference and future periods over the UHRB. (a) Time series of SSI-3 associated with reservoir inflow and release for the low-emission SSP126 scenario. Blue and orange intervals indicate their uncertainties, respectively. (b) Number of drought events for the reference period (1985–2014), near-future period (2031–2060), and far-future period (2071–2100). Colored bars are ensemble means and error bars represent the estimated difference in the number of drought events among the five GCMs. (c-d) is the same as (a-b), but for the medium-emission SSP370 scenario. (e-f) is the same as (a-b), but for the high-emission SSP585 scenario

26

**Figure 9.** Heat map representation of (a) drought duration and (b) drought severity for the GCM-averaged SSI-3 series. The symbols R1, R2 and R3 indicate the minimum, maximum, and mean values during the reference period (1985–2014). N1, N2 and N3 are the same, but for the near-future period (2031–2060). F1, F2, and F3 are for the far-future period (2071–2100). Additionally, SSP126-I and SSP126-R are associated with reservoir inflow and release in the SSP126 scenario, SSP370-I and SSP370-R with the SSP370 scenario, and SSP585-I and SSP585-R with the SSP585 scenario.

# 4.4 Adaptability of optimal operating policies to future hydrological droughts

Optimal reservoir operating policies can be explored as a potential means for human adaptation to future climate change. Previous studies have highlighted the promise of such policies in mitigating the adverse impacts of severe hydrological events (Wu et al., 2023; Sun et al., 2023; Yun et al., 2021b; Levey and Sankarasubramanian, 2025). However, this promising approach has largely remained conceptual, with limited practical validation to date. In this section, the NSGA-II algorithm was applied to derive 100 Pareto-optimal solutions based on historical inflow observations (Figure S5), and the implications of these solutions for future hydropower generation and drought characteristics under climate change were systematically examined.

The simulation results of these 100 optimal operating policies for hydropower and SSI-3 drought characteristics under future climate change conditions are then reported in Figure 10 using parallel-axis plots. The historically derived operating policy is outlined in black for comparison. These plots label each operating policy as a shaded line that intersects each vertical axis at the

achievable performance value, and the axes are oriented with the optimal direction upwards. The ideal policy in Figure 10 is, therefore, a horizontal line across the top of each axis. Nevertheless, these lines usually intersect between pairs of vertical axes because superior performance in one indicator comes at the cost of poorer performance in another. For instance, poor power generation guarantee rates inevitably have an impact on the goal of maximizing annual average hydropower generation. All optimal policies have similar future annual average hydropower generation, except for the far-future period under SSP126. They have a wide-spanning range of guarantee rates, such as 76.69%-84.32% for the near-future period under SSP126 and 61.07%-72.05% for the far-future period under SSP585. Additionally, as can be seen in all subplots of Figure 10, all the optimal operating policies result in more hydropower benefits but also a higher drought frequency than the historically derived policy. The SSI-3 series associated with optimal reservoir release is broken into more drought events where the average duration and severity of droughts don't change much. The most challenging drought management task remains in the future-period under SSP585, during which the historically derived policy has the relatively slightest drought severity. On the whole, a small number of optimal policies can achieve robust and satisfying levels of all considered indicators across plausible future scenarios, further revealing the potential for the application of optimal operating policies to short-term hydrological droughts.

al., 2023; Firoz et al., 2018).

**Figure 10.** Trade-offs among hydropower generation, guarantee rate, and SSI-3 drought characteristics under optimal and historical reservoir operating policies using parallel coordinates plots. Panels (a–b) correspond to the near-future (nf) and far-future (ff) under SSP126, (c–d) under SSP370, and (e–f) under SSP585. The grey lines represent Pareto-optimal policies, while the red and blue lines indicate the solutions with the highest guarantee rate and maximum hydropower generation, respectively, and the black line indicates the historical operating pattern. Each axis represents an objective, with the optimal direction oriented upwards.

Future development of drought-focused reservoir operation policies could incorporate a range of drought characteristics as direct optimization objectives. In the game of hydropower generation and drought resilience, there is still some room for improvement in drought mitigation in this study. It is also possible to use optimal operating policies along with other human actions, like inter-basin water transfers and urbanization, to prepare for potential future droughts (Wu et

### 5 Conclusions

By performing a simultaneous simulation of a LSTM-based reservoir inflow model and a physics-guided reservoir operation model, this study achieved a fully automated ML projection of river streamflow changes over the UHRB under different future scenarios and used it to project the associated hydrological drought. Climate change and reservoir operation were successively considered in the projections to reveal their different roles. Additionally, the trade-off between future hydrological droughts and operating benefits (i.e., hydropower generation and power generation guarantee rate) was investigated by optimizing the reservoir operating policies. The main findings are summarized as follows:

- 1. A reasonable LSTM-based model architecture is recommended for hydrological simulation in the reservoir-regulated region. If the historical meteorological simulation of ISIMIP3b is used instead of hydrological observations, it can still reflect the inflow and outflow of Ankang Reservoir as well as changes in reservoir storage. This demonstrates the feasibility of projecting future streamflow and associated hydrological droughts using ML approaches.
- 2. Future climate change over the UHRB tends to reduce natural streamflow and exacerbate hydrological droughts, especially in the far-future period (2071-2100) under the SSP585 scenario. While the operation of the Ankang Reservoir can mitigate the frequency, duration, and severity of short-term hydrological droughts (SSI-1 and SSI-3), it shows limited effectiveness in alleviating long-term droughts (SSI-12).
- 3. Optimal reservoir operating policies at Ankang Reservoir, designed to maximize hydropower generation and power generation guarantee rates, can effectively reconcile the trade-offs between hydrological drought and hydropower benefits, especially in the near-future period (2031-2060). Compared to the historically derived policy, these optimal strategies yield higher

hydropower benefits but may also lead to increased drought frequency. The finding that a small subset of optimal policies can consistently achieve robust and satisfactory performance across all evaluated indicators under plausible climate scenarios underscores their potential in enhancing regional water resource management under climate change.

### **Declaration of Competing Interest**

The authors declare that they have no known competing financial interests or personal relationships that could have appeared to influence the work reported in this paper.

### Data availability

The code that supports the findings of this study is available from the corresponding author upon reasonable request. The ISIMIP3b data used in producing this paper are available at https://data.isimip.org/search/tree/ISIMIP3b/InputData/. Observed streamflow data are available from the Bureau of Hydrology of the Yangtze Water Resources Commission of China (https://www.cjh.com.cn).

## **Author contribution**

YG, KC and SH designed the study. SH, SS, and YL developed the models, with SH and LZ implementing them. SH drafted the manuscript in close collaboration with YG, SS, YL contributed to the data curation. Throughout the study period, all the authors engaged in discussions regarding the results, provided critical feedback, and approved the final version of the paper.

### Supplement

The supplement related to this article is available.

581

## Financial support

- This research has been supported by the National Key Research and Development Program
- of China (2023YFC3209502) and National Natural Science Foundation of China (U2340217 and
- 42577102).

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
