# Peer review of "Projections of future hydrologic drought in a reservoir-regulated region: the role of"

_EGUsphere, 2025_

## Author Comment (AC2)

**Reply to Reviewers' comments (Reviewer#2)**

**Legend**

Reviewers' comments

Authors' responses

Direct quotes from the revised manuscript

**Reviewer #2:**

The manuscript investigates how climate change and reservoir operation jointly shape future hydrological droughts in a heavily regulated basin, using an Ankang Reservoir in the upper Hanjiang River Basin in China as an example. A hybrid framework was developed by coupling a LSTM-based hydrological model with a physics-guided LSTM reservoir operation module, which was then driven by ISIMIP3b CMIP6 projections (five GCMs, three SSPs) to simulate regulated streamflow in the near- (2031-2060) and far-future (2071-2100) periods. Hydrological drought characteristics are quantified using SSI-1, SSI-3, and SSI-12, and a multi-objective optimization (NSGA-II with RBF parameterization) is used to explore whether optimal operating policies can balance hydropower benefits and drought risks.

Overall, the paper addresses an important and timely topic, and the methodology is sound and clearly implemented. I believe the manuscript is suitable for publication after minor revisions to clarify the novelty, add some methodological details, and strengthen the discussion of limitations and applicability.

Response: Thank you for the positive and constructive assessment. We appreciate your recognition that the topic is timely and that the proposed methodology is sound and clearly implemented. In response to your suggestions, we have revised the manuscript accordingly by clarifying the novelty, adding key methodological details, and strengthening the Discussion on limitations and applicability (including a new Section 4.5). We believe these changes improve the clarity and robustness of the study.

Major comments

1. Clarify the paper's novelty

The Introduction already reviews several related works (e.g., VIC-Reservoir and CSSPV2+reservoir frameworks for future drought projections, and recent studies coupling

reservoir models with CMIP6). However, the specific advances of this study relative to those works could be articulated more explicitly.

Please more clearly state what is new in this manuscript: e.g., (i) replacing traditional hydrological models with a fully data-driven LSTM for inflow and outflow, and (ii) explicitly combining this hybrid model with multi-objective optimization of operating policies to examine drought–hydropower trade-offs under future climate scenarios.

It would help if the end of the Introduction contained a short, itemized list of the main contributions to distinguish this work from previous hybrid and reservoir–drought studies.

Response: Thank you for this constructive and insightful comment. We have substantially revised the final part of the Introduction to clearly articulate the novelty of this manuscript. Specifically, we now explicitly state that the main contributions of this study are:

(i) replacing traditional process-based hydrological models with a fully data-driven LSTM framework to quantify hydrological drought by directly simulating reservoir inflow and outflow dynamics; and, (ii) explicitly exploring the adaptive performance of optimal operating policies under future climate change scenarios.

In addition, we have restructured the final paragraph of the Introduction to present these contributions in a concise and itemized manner, thereby clearly distinguishing this work from previous hybrid and reservoir–drought studies based on process-based models or conceptual operating rules. These revisions have been incorporated in Lines 121–136 of the revised manuscript.

2. Assumption of stationary operation policy when projecting future droughts

The study assumes that the historical operation policy learned by the physics-guided LSTM (1992–2020) is directly applicable to the reference and future climate periods (ISIMIP3b_ref and ISIMIP3b_fut experiments). This is a reasonable and often necessary assumption, but it should be discussed more explicitly as a limitation: operation rules in reality may adapt to changing demands, policies, or infrastructure.

Response: Thank you for this insightful comment. We agree that our future projections rely on the assumption that the historical reservoir operating policy learned by the physics-guided LSTM module remains applicable to the reference and future periods. Following your suggestion, we have now explicitly acknowledged this as a key limitation in the Discussion (Section 4.5, Lines 569–580). Specifically, we clarify that the projected drought responses under reservoir regulation should be interpreted as conditional on a "stationary operating policy" hypothesis, rather than representing fully adaptive future management outcomes. The revised text is also shown below.

**I) Non-stationarity of the operating environment and implications for framework applicability:** in our framework, the physics-guided LSTM module learns reservoir operating behavior from historical conditions and is subsequently applied to the reference and future periods. This procedure implicitly assumes that the learned decision logic remains transferable under changing climate regimes. Notably, our previous analyses indicate that, for the case investigated here, the reservoir-regulated hydrological response exhibits relatively stable and consistent patterns over multi-decadal timescales, supporting the feasibility of using a surrogate model to represent operational behavior (He et al., 2023). However, such consistency is case-specific and may not hold in other basins or under intensified human interventions, which could progressively undermine the stationarity assumption. Therefore, the projected drought responses under reservoir regulation in this study should be interpreted as conditional on a "stationary operating policy" hypothesis, rather than as fully adaptive future management outcomes.

3. Uncertainty and generality of the results

The study uses five ISIMIP3b GCMs and a single basin–reservoir case. While this is already a substantial effort, readers would benefit from a more explicit reflection on the scope and limitations. Please expand the discussion of uncertainties related to (i) the limited number of climate models, (ii) the single-case setting (UHRB, Ankang) and whether the conclusions are transferable to other types of reservoirs or climate regimes, (iii) A short paragraph in the Discussion or Conclusions explicitly addressing "limitations and future work" would strengthen the paper and guide follow-up studies.

Response: Thank you for the comment. Instead of adding this information in the Discussion/Conclusions, we have strengthened the justification of the climate forcing in the Data section (Section 2.2) by citing multiple studies demonstrating the robustness of ISIMIP3b across many regions of China (Lines 189–190; Kang et al., 2023; Yun et al., 2021a; He et al., 2023). This provides explicit evidence supporting our choice of the ISIMIP3b ensemble for future projections.

References:

He, S., Chen, K., Liu, Z., and Deng, L.: Exploring the impacts of climate change and human activities on future runoff variations at the seasonal scale, J Hydrol, 619, 129382, 10.1016/j.jhydrol.2023.129382, 2023.

Kang, S., Yin, J., Gu, L., Yang, Y., Liu, D., and Slater, L.: Observation‑constrained projection of flood risks and socioeconomic exposure in China, Earth's Future, 11, e2022EF003308, 10.1029/2022ef003308, 2023.

Yun, X., Tang, Q., Sun, S., and Wang, J.: Reducing Climate Change Induced Flood at the Cost of Hydropower in the Lancang‑Mekong River Basin, Geophysical Research Letters, 48, e2021GL094243, 10.1029/2021gl094243, 2021a.

Minor comments

1. In the Abstract, there seems to be a small typo in "pow generation guarantee rate"; please correct to "power generation guarantee rate".

Response: Thank you for pointing this out. We have corrected the typo in the Abstract.

2. Consider defining "LSTM+Reservoir" more explicitly at its first appearance in the Abstract or Introduction (e.g., "a hybrid LSTM-based hydrological and physics-guided reservoir operation model").

Response: Thank you for this helpful suggestion. To avoid potential misunderstanding at an early stage of the manuscript, we have removed the term "LSTM+Reservoir" from both the Abstract and the Introduction, and instead provide its full definition and description in the experimental design section (Section 3.3), where the modeling framework is introduced in detail. The revised Abstract has been updated in Lines 26–29 (also shown below).

A long and short-term memory (LSTM)-based hydrological model, coupled with a physics-informed LSTM reservoir model, is developed and driven by bias-corrected climate outputs from five global climate models to project future drought conditions under three scenarios (SSP126, SSP370, and SSP585).

3. Notation and acronyms. Please ensure that all acronyms are defined at first use (e.g., SSI, NBS, NSGA-II, RBF). Some are introduced in the text or caption but could be clarified earlier for readability.

4. It may help to add a short list of symbols for key variables (e.g., V, I, O, THP, PGR, D, S) either in the main text or Supplement.

Response: Thank you for these helpful suggestions. We have revised the manuscript to define all acronyms at their first occurrence. We have also added a list of symbols for the key variables in the Supplementary Material to improve clarity. It is also shown below.

Acronyms and Notation

For ease of reading, all important and notations in the main text are summarized below.

**Acronyms**

| | |
|---|---|
| UHRB | Upper Hanjiang River Basin |
| SSI | Standard Streamflow Index |
| LSTM | Long and Short-Term Memory |
| GCM | Global Climate Model |
| SSP | Shared Socioeconomic Pathway |
| ML | Machine Learning |
| CMA | China Meteorological Administration |
| ISIMIP3b | Inter-sectoral Impact Model Intercomparison Project 3b |
| NSGA-III | Nondominated sorting genetic algorithm version III |
| FDC | Flow Duration Curve |
| NSE | Nash–Sutcliffe efficiency |

**Notation for key variables**

| | |
|---|---|
| $f$ | Operating objective vector |
| $\pi_\theta^*$ | Optimal operating policies |
| $w^H$ | Historical climate conditions |
| $V$ | Reservoir storage |
| $I$ | Reservoir inflow |
| $O$ | Reservoir outflow |
| $D$ | Duration of hydrological drought |
| $S$ | Severity of hydrological drought |

5. For the parallel coordinate plots in Figure 10, you might consider adding an arrow in the left to clarify which direction is "better" for each metric (e.g., upward is optimal for all axes) to help non-expert readers interpret the trade-offs.

Response: Thank you for your suggestion. We note that all axes are oriented in the same "better upward" direction, so an explicit arrow is not strictly necessary. Instead, to improve readability for non-expert readers, we have clarified the direction of improvement in the caption of Figure 10 by adding the statement: "Each axis represents an objective, with the optimal direction oriented upwards." This annotation helps readers interpret the trade-offs shown in the parallel coordinate plots.